# Host Cell Calpains Can Cleave Structural Proteins from the Enterovirus Polyprotein

**DOI:** 10.3390/v11121106

**Published:** 2019-11-28

**Authors:** Mira Laajala, Minna M. Hankaniemi, Juha A. E. Määttä, Vesa P. Hytönen, Olli H. Laitinen, Varpu Marjomäki

**Affiliations:** 1Department of Biological and Environmental Science/Nanoscience Center, University of Jyväskylä, Survontie 9C, FI-40500 Jyväskylä, Finland; mira.a.laajala@jyu.fi; 2Faculty of Medicine and Health Technology, Tampere University, FI-33014 Tampere, Finland; minna.hankaniemi@tuni.fi (M.M.H.); juha.maatta@tuni.fi (J.A.E.M.); vesa.hytonen@tuni.fi (V.P.H.); olli.laitinen@tuni.fi (O.H.L.); 3Department of Clinical Chemistry, Fimlab Laboratories, Pirkanmaa Hospital District, FI-33520 Tampere, Finland

**Keywords:** enterovirus, calpain, proteolytic processing, RNA virus, polyprotein

## Abstract

Enteroviruses are small RNA viruses that cause diseases with various symptoms ranging from mild to severe. Enterovirus proteins are translated as a single polyprotein, which is cleaved by viral proteases to release capsid and nonstructural proteins. Here, we show that also cellular calpains have a potential role in the processing of the enteroviral polyprotein. Using purified calpains 1 and 2 in an in vitro assay, we show that addition of calpains leads to an increase in the release of VP1 and VP3 capsid proteins from P1 of enterovirus B species, detected by western blotting. This was prevented with a calpain inhibitor and was dependent on optimal calcium concentration, especially for calpain 2. In addition, calpain cleavage at the VP3-VP1 interface was supported by a competition assay using a peptide containing the VP3-VP1 cleavage site. Moreover, a mass spectrometry analysis showed that calpains can cleave this same peptide at the VP3-VP1 interface, the cutting site being two amino acids aside from 3C’s cutting site. Furthermore, we show that calpains cannot cleave between P1 and 2A. In conclusion, we show that cellular proteases, calpains, can cleave structural proteins from enterovirus polyprotein in vitro. Whether they assist polyprotein processing in infected cells remains to be shown.

## 1. Introduction

Enteroviruses belong to a large family on non-enveloped RNA-viruses Picornaviridae. Enteroviruses cause diverse diseases in humans with symptoms varying from mild, such as common cold, to more severe such as meningitis, encephalitis, and myocarditis [1,2,3,4]. In addition to acute diseases, enteroviruses have been associated with chronic diseases like type I diabetes or asthma [5,6,7]. However, there is no antiviral on the market against enteroviruses. Our previous studies have shown that host cell proteases, calpains, have a role in enterovirus infection after entry and RNA release and that calpain activity increased during infection at around 3 h post infection (p.i.), when translation/replication starts to accelerate [8,9].

Enteroviruses are small viruses with single-stranded positive-sense 7500 nt RNA genome inside an icosahedral capsid. After cell entry, the viral RNA is released into the host cytoplasm, where it is directly translated to a single polyprotein and acts as a template in RNA synthesis. The four structural proteins (VP1–VP4) are encoded by P1 region in the polyprotein, whereas P2 and P3 regions contain several nonstructural viral proteins that either directly participate in viral replication or remodel the host cell architecture to promote the viral replication [10]. According to current knowledge, the polyprotein is considered to contain all the necessary proteins needed for polyprotein processing as the nonstructural proteins also include viral proteases 2A and 3C and a precursor protein 3CD. These viral proteases have been shown to cleave the polyprotein co- and posttranslationally in order to release the individual proteins [11]. The viral protease 2A is located immediately after the P1 region and has been shown to carry out the first cleavage of itself and the rest of the polyproteins (P2–P3) out from the P1 region already during translation [10,11,12,13]. In addition, in trans-polyprotein cleavage by 2A has been reported [14]. The other cleavages, with the exception of capsid maturation cleavage VP0 to VP4 and VP2, have been shown to be carried out either by 3C or 3CD precursor, which are located in the P3 region [11,15]. In addition to the polyprotein cleavage, the viral proteases have been shown to cleave several host cell proteins in order to promote infection and to contribute the pathogenicity [10]. The identified cell targets include, for example, proteins involved in cellular transcription, translation, and immune responses [10,16]. The viral proteases 2A and 3C have cysteine in their catalytically active site, being structurally similar to chymotrypsin-like serine proteases [17]. Inside the enterovirus genus, the catalytic cysteine of these proteases and other amino acids in the catalytic triad are fully conserved [10].

Calpains are papain-like cysteine proteases found in almost all eukaryotes [18]. There are 15 different calpains in humans, and the first identified was the conventional calpains 1 and 2. To form a functional protease, the catalytic subunit (80 kDa) of calpain 1 and 2 form a heterodimer with a regulatory subunit CAPNS1 (30 kDa), which is common to both conventional calpains [18]. Calpains exist in the cytoplasm as proenzymes and are activated when intracellular Ca^2+^ levels are increased. Instead of complete degradation, calpains cause limited proteolysis mainly in the inter-domain unstructured regions of their substrates. It has been suggested that this is due to a narrow and deep active site in the CysPc protease domain [19]. In order to fit into the active-site cleft, the substrate proteins have to be in an extended conformation. Several protein targets have been found for calpains in in vitro studies such as cytoskeletal proteins, membrane associated proteins, transcriptional factors, as well as kinases and phosphatases [9,20]. However, the substrate specificity of calpains has long been unclear, and it has been suggested that the 3D structure of the target protein may play more important role in substrate recognition compared to amino acid sequence of the target [21,22].

Calpains have been associated with enterovirus infection already in earlier studies: they have been suggested to have a role in coxsackie B virus-induced vesicle trafficking inside cells, necrosis, and autophagy [23,24,25]. Despite earlier studies by us and others, mechanistic understanding of calpain action during the replication/translation phase of enterovirus infection has remained unclear. In addition, the involvement of host cell proteases to take part in proteolytic processing has remained a possibility, but none have been proven so far. Here, we show that calpains are able to proteolytically process the enteroviral P1 region of enterovirus B species at two specific sites in vitro, thus suggesting that they may contribute to efficient capsid maturation of enteroviruses.

## 2. Methods

### 2.1. Reagents

Purified calpain 1 (human) and 2 (rat) proteases were from Calbiochem. Purified viral proteases 2A and 3C were expressed in *E. coli* as described before [26]. Calpain inhibitor I (N-Acetyl-Leu-Leu-norleucinal) was from Roche (Basel, Switzerland). Elastatinal was from Santa Cruz biotechnology (Dallas, Texas, USA). Acetonitrile (ACN), formic acid (FA), water (UHPLC-MS grade), triethylammonium bicarbonate buffer 1 M (TEAB), sodium dodecyl sulfate (SDS), iodoacetamide (IAA), trifluoroacetic acid (TFA), ammonium bicarbonate (ABC), and urea were all purchased from Sigma Aldrich Corp. (St. Louis, MO, USA). Sample clean up tips (C18) were from Thermo Fisher Scientific (San Jose, CA, USA). A kit (Bio-Rad DC) and bovine serum albumin standard were purchased from Bio-Rad Laboratories Inc. (Hercules, CA, USA), and 30 kDa molecular weight cut-off (MWCO) centrifugal devices were from PALL (Port Washington, NY, USA).

### 2.2. Cells

Human alveolar basal epithelial A549 cells (ATCC) were cultured at 37 °C in Dulbecco’s modified Eagle’s medium (DMEM, Invitrogen, Carlsbad, CA, USA)) containing 10% fetal bovine serum (Invitrogen), 1% glutamax (Invitrogen), and 1% penicillin and streptomycin antibiotics (Invitrogen). Sf9 cells (Invitrogen) were cultured in insect-XPRESS culture medium (Lonza, Basel Switzerland) and maintained in non-humidified incubator at 27 °C, where the culture flask was rocked at 120 rpm. The cultures where passaged in mid-log phase between 4 × 10^6^–6 × 10^6^ cells/mL with the seeding density of 1 × 10^6^ cells/mL.

### 2.3. Production of P1 and P1-2A* Constructs

Two baculoviral transfer vectors (pOET5) containing expression cassettes under polyhedrin promoter were ordered from GeneArt (Regensburg, Germany). pOET5-CVB1-P1 vector containing expression cassette for CVB1 capsid proteins VP1–4 and pOET5-CVB1-P1-2A_C>A_ vector containing expression cassette for CVB1 capsid proteins VP1–4 and 2A protease with cysteine-to-alanine substitution (resulting in the loss of protease function) were utilized. CVB1 field isolate [7] was used as a template for these constructs. The recombinant baculoviruses were produced according to the FlashBAC baculovirus expression system (Oxford Expression Technologies, Oxford, UK). In FlashBAC ULTRA baculovirus genome, the genes coding for chitinase, cathepsin, p10, p26, and p74 are deleted. CVB1-P1 (P1) and CVB1-P1-2A_C>A_ (P1-2A*) polyproteins were produced in Sf9 insect cells, the cells containing the polyproteins were harvested 3–6 dpi by centrifugation, and proteins were released from the cells by freezing and thawing the cells.

### 2.4. Calpain In Vitro Cleavage Assays

A reaction mixture containing the following components was prepared: 1.6 μg of P1 or P1-2A* containing lysate, 1 U calpain proteases, 2 mM CaCl_2_, and PBS. In the inhibitor assay, the reaction mixture also contained 200 μM calpain inhibitor I or 250 μM elastatinal.

In the calpain titration assay, the reaction mixture contained 1.6 μg of P1 containing lysate and 0.01 U, 0.1 U, 0.5 U, or 1 U of calpain proteases in PBS. The mixture also contained 2 mM CaCl_2_ and 4 mM EGTA when indicated. In the calcium titration assay, the reaction mixture contained 1.6 μg of P1 containing lysate; 1 U of calpain proteases; and 0, 0.002, 0.02, 0.2, or 2 mM CaCl_2_ in PBS.

The reactions were incubated at 25 °C in water bath for 2 h. The reactions were terminated by adding 4× SDS-PAGE sample buffer containing mercaptoethanol (1× final concentration).

### 2.5. In Vitro Cleavage Assay with Viral Proteases

The reaction mixture contained 1.6 μg of P1 or P1-2A* containing lysate, 750 ng of purified viral proteases 2A or 3C in buffer containing 20 mM HEPES (pH 7.4), 120 mM KCH_3_COO, 4 mM Mg(CH_3_COO)_2_, and 5 mM DTT.

In the inhibitor assay, the viral proteases were incubated with A549 cell homogenate, which was prepared as described [27]. The reaction mixture contained 75 μg of the homogenate, 375 ng of 2A or 3C with or without 200 μM calpain inhibitor I or 250 μM elastatinal, and reaction buffer (20 mM HEPES (pH7.4), 120 mM KCH_3_COO, 4 mM Mg(CH_3_COO)_2_, and 5 mM DTT).

The reactions were incubated at room temperature (+22 °C) for 18 h, after which 4× SDS-PAGE sample buffer containing mercaptoethanol (1× final concentration) was added to terminate the reaction.

### 2.6. Infection Assay in Cells

A549 cells were infected with EV1 (Farouk strain, ATCC), CVB3 (Nancy strain, ATCC), or CVB1 (Conn 5 strain, ATCC) with 2 × 10^8^ PFU/mL. The viruses were first bound on ice for 1 h in DMEM supplemented with 1% fetal bovine serum and 1% glutamax. Next, excess virus was washed away with PBS, and fresh DMEM supplemented with 1% fetal bovine serum and 1% glutamax containing 200 μM calpain inhibitor I or 250 μM elastatinal was added. The infection was then allowed to proceed at 37 °C for 5.5 h. Finally, the cells were either fixed with 4% paraformaldehyde for 30 min or collected into 2× Laemmli buffer containing mercaptoethanol (1× final concentration).

### 2.7. Immunofluorescence Labeling and Microscopy

The cells were permeabilized with 0.2% Triton X-100 and then treated with primary antibodies: rabbit anti-EV1 [28] for EV1 or monoclonal mouse Enterovirus clone 5-D8/1 antibody (Dako) for CVB3 and CVB1. After 1 h of incubation, excess primary antibody was washed with PBS and cells were treated with secondary antibodies: goat anti-rabbit Alexa 555 (Invitrogen) or goat anti-mouse Alexa 555 (Invitrogen). Secondary antibody was washed with PBS and nuclei stained with DAPI in PBS. Finally, the cells were mounted into mowiol-DABCO.

Samples were imaged with Olympus FV1000-IX81 confocal microscope using 543-nm HeNe and 405-nm diode lasers. The imaging was carried out with 60× UPLSAPO objective (NA 1.35), and levels for the laser power and detector amplification were optimized using negative infection control.

### 2.8. Western Blot

Samples were first boiled and then run in 4–20% Miniprotean TGX gradient gel (Bio-Rad Laboratories Inc.). Next, the proteins were transferred to PVDF membrane (Immobilon-P, Merck Millipore; Burlington, MA, USA), after which the blot was blocked overnight with 5% BSA in 0.05% tween in TBS. Poly A binding protein (PABP) was detected using 1:100 dilution of mouse anti-PABP (Santa Cruz). Ras GTPase-activating protein-binding protein 1 (G3BP1) was detected using 1:1000 dilution of mouse anti-G3BP1 (Santa Cruz). VP3 was detected using in-house produced rabbit-anti-CVB1-6 polyclonal antibody. VP1, P1, and P1-2A* were detected using 1:4000 dilution of monoclonal mouse enterovirus clone 5-D8/1 antibody (Agilent Dako, Santa Clara, CA, USA). Gamma-tubulin was detected using 1:20,000 dilution of rat anti-tubulin antibody (Abcam, Cambridge, Unitied Kingdom). The primary antibodies were detected using corresponding horseradish peroxidase-conjugated secondary antibodies (cell signaling). Next, chemiluminescent substrate (supersignal West Pico PLUS, Thermo Fisher Scientific, Waltham, MA, USA) was incubated for 5 min and the chemiluminescence was detected using Chemidoc MP (Bio-Rad Laboratories Inc.). In the Figure 2, the total protein amount was revealed using stain-free method by UV-activating the gel with Chemidoc MP (Bio-Rad Laboratories Inc.) before transfer to PVDF membrane. The western blot results were quantified using the gel analyzer tool embedded in Image J.

### 2.9. Cleavage Assay with the Peptide

A 30-amino-acid peptide from CVB1 VP3 and VP1 capsid protein interface (peptide sequence: NH2-MLKDTPFIRQDNFYQ*GPVEESVERAMVRVA-COOH; putative 3C cleavage site indicated by asterisk) was chemically synthesized (Proteogenix, Sciltigheim, France) and treated with calpain 1, calpain 2, and 3C proteases. Two micrograms of the peptide was incubated in the presence of 0.25 U calpain 1, 0.25 U calpain 2, or 750 ng 3C protease. Reactions containing calpain 1 or calpain 2 were incubated two hours in 25 °C water bath in PBS containing 2 mM CaCl_2_. 3C protease reactions were incubated 20 h at room temperature in 20 mM Tris pH 7.3, 100 mM NaCl, and 1 mM EDTA.

In the competition assay, a reaction mixture containing the following components was prepared: 1.6 μg of P1 containing lysate, 1 U calpain proteases 1 or 2, or 750 ng of 3C protease with or without 100 μg of the peptide. The calpain reactions were incubated in 25 °C water bath for 2 h in PBS containing 2 mM CaCl_2_. 3C protease reactions were incubated for 17 h at 22 °C in 50 mM Tris pH 7.5, 150 mM NaCl, 1 mM EDTA, and 1 mM DTT. The reactions were terminated by adding 4× sample buffer containing mercaptoethanol (1× final concentration).

### 2.10. Mass Spectrometry

Cleavage assay’s peptides with or without proteases were analyzed with Eksigent 425 NanoLC coupled with Sciex high speed TripleTOF™ 5600+ mass spectrometer using sequential window acquisition of all theoretical mass spectra (SWATH-MS) proteomics. Prior to SWATH analysis method, a relative protein quantification library was built from several data-dependent analysis (DDA) runs with the same LC gradient and instrument settings that were used for SWATH analyses. Before MS runs, samples were subjected to reduction, alkylation, and digestion with calpain protease instead of trypsin as described in detail in Reference [29]. After digestion, peptides were diluted to 14 µl of sample buffer (2% acetonitrile and 0.1% formic acid) and 1 µl of sample was injected to the triple TOF mass spectrometry. Library was created using Protein Pilot 4.7 (Sciex, Redwood City, CA, USA), and all DDA run spectra were identified against coxsackievirus B3 and calpain (human and rat) UniprotKB/SwissProt protein library added with synthetic CVB1 VP3-VP1 peptide. Quantification was done by Peak Viewer and Marker Viewer (Sciex).

### 2.11. Statistical Testing

The statistical analysis was carried out using GraphPad Prism software. (GraphPad Software, San Diego, CA, USA) The data are presented as means ± SEM. The statistical significance was determined using parametric one-way ANOVA with Bonferroni’s multiple comparisons test.

## 3. Results

### 3.1. Calpain Inhibitor Prevents the Enterovirus Infection Whereas 2A Inhibitor Does Not

The effect of calpain inhibitor on the infection of A549 cells with different serotypes of enterovirus B species was tested, and infection was detected by the marked accumulation of VP1 in the cytoplasm by immunofluorescence. The results showed that the presence of 200 µM calpain inhibitor strongly reduced the VP1 signal and decreased the number of EV1-, CVB3-, and CVB1-infected cells compared to the control infection (Figure 1A). The input virus used in the infection was seen as bright spots in endosomes, particularly in CVB3 infected cells (Figure 1A). In contrast, the addition of 250 µM elastatinal, which has been shown to inhibit 2A [30], did not suppress the infection of A549 cells by any of these viruses (Figure 1A). The same observation was also evident when the levels of VP1 protein were analyzed from the whole cell population using western blotting (Figure 1B).

### 3.2. Calpain Inhibitor Cross-Reacts with Viral Proteases 2A and 3C

Both calpains and viral proteases of enteroviruses have cysteine in their catalytically active site. Because our present and earlier results [8] showed that the enterovirus infection was inhibited by a calpain inhibitor, we also wanted to test whether the inhibitor has an effect on viral proteases. We carried out an in vitro assay, where A549 cell homogenate was incubated with purified viral proteases with or without a calpain inhibitor (Figure 2). The cleavage action of 2A and 3C was evaluated by detecting the known host cell targets of the proteases, poly A binding protein (PABP), and Ras GTPase-activating protein-binding protein 1 (G3BP1). Western blot results showed that the calpain inhibitor prevented the action of both proteases: the amount of PABP protein was in the same level as in controls lacking proteases, and the amount of G3BP1 cleavage product was clearly less in the presence of the inhibitor than without it (Figure 2). In addition, we confirmed that another protease inhibitor, elastatinal, which has been shown to inhibit 2A, inhibited the action of 2A but not 3C (Figure 2.).

### 3.3. Calpains Can Cleave P1 into VP1 and VP3 In Vitro but Cannot Release P1 from the Polyprotein

Because we noticed that calpain inhibitor also cross-reacts with viral proteases, we wanted to study the action of calpains and viral proteases separately. We used the baculovirus-infected Sf9 insect cells to produce the P1 region of CVB1 polyprotein (Figure 3A). In addition, since our preliminary study showed that the 2A inhibitor elastatinal could not prevent virus infection, we also produced a construct where the P1 region was followed by mutated 2A (P1-2A*) in order to study the cleavage site between P1 and 2A (Figure 3A). Viral protease 2A cotranslationally cleaves between itself and P1 region [11,12,13], so in order to avoid this cleavage action by 2A, the active site cysteine of 2A (C110) was mutated into alanine.

Purified calpain proteases 1 and 2 as well as viral proteases 2A and 3C were incubated with P1 or P1-2A* in vitro, after which the cleavage products were detected by western blotting. The results showed that the amount of P1 and P1-2A* decreased in the presence of both calpain 1 and 2 enzymes (Figure 3B,C). More importantly, when the P1 construct was treated with calpains, the cleavage products corresponding to VP1 and VP3 proteins appeared and accumulated compared to the control, where no calpains were added (Figure 3B,D,E). The amounts of both VP1 and VP3 were statistically higher compared to control, when P1 was treated with calpain 2 (*p* < 0.01 and 0.001 for VP1 and VP3, respectively). Also, calpain 1 produced more VP1 and especially VP3 compared to control, but the results were not statistically significant. This result thus suggested that calpain 2 is more potent at cleaving the P1 region in vitro as compared to calpain 1. As we saw, a low background signal of VP1 in the insect cell produced a P1-containing sample; we tested whether this may be due to a low activity of endogenous proteases, perhaps close to mammalian calpains. Addition of the calpain inhibitor after day 1 in the infected insect cell culture prevented the accumulation of VP1 band at day 2, suggesting that the inhibitor prevented proteolytic processing carried out by endogenous calpain or a close relative in the insect cells (Mira Laajala, University of Jyväskylä, Finland, Unpublished experiments performed on insect cell cultures). Understandably, we could not add calpain inhibitor in our assays when we assessed the proteolytic processing of exogenously added calpains, which caused the presence of a low endogenous amount of processed VP1 in the assays (Figure 3, Figure 4 and Figure 5B). As a positive control, we used purified viral protease 3C, which resulted in a statistically significant increase of released VP1 and VP3 proteins compared to the negative control (Figure 3B,D,E) (*p* < 0.01 and 0.001 for VP1 and VP3, respectively). Additionally, our results showed that 2A viral protease cannot cleave at the P1-2A cleavage site in trans, when it is not attached to the polyprotein (Figure 3B,D,E). It was also evident that calpains cannot cleave at this cleavage site, since no cleavage product corresponding to either P1 or VP1 appeared (Figure 3B). Still, calpains were able to release VP3 from P1-2A*, confirming that the sites between VP3-VP1 and VP3-VP0 can be processed by calpains (Figure 3B,E). These results suggest that calpains, similar to 3C, can cleave P1 polyprotein of enterovirus B species in vitro and can produce VP1 and VP3 as cleavage products.

### 3.4. Calpain Activity is Concentration and Calcium Dependent and Can Be Prevented with Calpain Inhibitor In Vitro

Next, we titrated the amount of calpain proteases in an in vitro assay with P1 construct and 2 mM CaCl_2_. The results showed that VP1 was formed when 0.5 or 1 unit of calpain 1 or calpain 2 was used, respectively (Figure 4A). The processing was first seen as the appearance of a VP1+VP3 band, the intensity of which decreased as the amount of calpains was increased (Figure 4A). Interestingly, the VP1+VP3 band was already evident in the control sample, suggesting that host cell protease action in insect cell lysates could already promote the first cut within this in vitro reaction during 2 h at +25 °C. The appearance of VP1+VP3 band in the control sample was dependent on Ca^2+^, since the band did not appear in the absence of Ca^2+^ (Figure 4A), suggesting that the cleavage was carried out by calcium-dependent calpain protease. In addition, the action of calpain 2 (1 U) was blocked when the reaction was either done without 2 mM CaCl_2_ or when the CaCl_2_ was chelated with 4 mM EGTA (Figure 4A). On the other hand, the cleavage action of calpain 1 was not affected when Ca^2+^ was not added, suggesting that the amount of Ca^2+^ in insect cell-derived P1 lysate was enough for the activity of this protease (Figure 4A). Also, surprisingly, the addition of EGTA did not prevent the cleavage action of calpain 1 leading to VP1 (Figure 4A). In contrast, EGTA addition led to the almost complete disappearance of the P1 band. What is behind this finding remains to be studied further. The difference of Ca^2+^ dependency for calpain 1 and 2 was further shown in an in vitro assay, where the amount of CaCl_2_ was titrated against 1 unit of calpain 1 or 2 (Figure 4B). The results showed that calpain 2 released VP1 from P1 only with Ca^2+^ concentrations higher than 0.2 mM while calpain 1 released VP1 even without additional CaCl_2_.

Since both calpains 1 and 2 cleaved the viral P1 region, we wanted to verify whether the cleavage action of calpains can be prevented by carrying out an in vitro reaction, where purified calpains were incubated with the P1 polypeptide in the presence of calpain inhibitor. As above, the calpains without the presence of inhibitor produced cleavage products corresponding to VP1 and VP3, whereas in the presence of inhibitor, these cleavage products were not formed (Figure 4C,D). Altogether, these results further confirmed that the appearance of the bands corresponding to VP1 and VP3 proteins was due to action of calpains.

### 3.5. Calpains Cleave Specifically at the VP1-VP3 Cleavage Site

Since we observed that calpains cleave P1 and produce VP1 and VP3 proteins, we concluded that calpains can cleave at around the site between VP0-VP3 and VP3-VP1. Since the sequence motif that is cut by calpains may differ from the viral proteases, we used synthetic peptide covering the VP3-VP1 cleavage site to study the cleavage site of calpains in more detail. This 30-amino-acid peptide (NH2-MLKDTPFIRQDNFYQ*GPVEESVERAMVRVA-COOH) also included the putative cleavage site of viral protease 3C, indicated by asterisk. Unfortunately, a proper peptide corresponding to the cleavage site of VP0-VP3 could not be generated because of its low water solubility.

The peptide was used in mass spectrometry analysis to reveal the exact cleavage site for calpains. The peptide was incubated with purified calpains or 3C, and the cleavage site was revealed by analyzing the cleavage products produced by the proteases using SWATH-MS (Figure 5A). Based on the cleavage products (sequences) with the highest intensities, the results showed that the synthetic peptide was cleaved by calpains 1 and 2 from (NH2–) MLKDTPFIRQDNF/YQGPVEESVERAMVRVA (–COOH), whereas 3C cleaved this peptide within its reported cleavage site (NH2–) MLKDTPFIRQDNFYQ/GPVEESVERAMVRVA (–COOH) (Figure 5A).

Furthermore, we utilized this peptide in a competition assay to study whether the action of calpains and 3C can be inhibited in its excess. Purified calpains or 3C protease were incubated with P1 construct in the presence or absence of 100 µg of the competing peptide. Both calpains 1 and 2 as well as 3C protease produced more VP1 in the absence of the peptide compared to the experiment in the presence of the peptide (Figure 5B), which suggests that these proteases cleave P1 at the VP3-VP1 cleavage site present in the competing peptide (Figure 5B). In conclusion, calpains 1 and 2 can specifically cleave between VP1 and VP3 and the cleavage site is two amino acids apart from the cleavage site used by 3C viral protease.

## 4. Discussion

Earlier, we showed that calpain proteases of the host cell participate in enterovirus infection after entry and RNA release [8]. That study pinpointed the effect of calpains in the early translation/replication phase at 3 h p.i. when infection induces calpain activity, and it was the same time window during which the virus replication could be blocked with calpain inhibitors [8]. However, the molecular mechanism of calpain action during enterovirus infection remained unclear. Here, we show using in vitro assays that calcium-dependent neutral calpain proteases can cleave the P1 polyprotein of enterovirus B species and, as a result, produce VP1 and VP3 capsid proteins detected by western blotting. Moreover, the cleavage specificity of calpains at the VP3-VP1 interface was confirmed with mass spectrometry analysis and competition assay with a peptide, including the VP3-VP1 cleavage site. On the other hand, we show that calpains are not able to make the cleavage between P1 and 2A. Our results thus show that, in addition to viral proteases, the P1 can be cleaved by host cell calpain proteases and that this processing is specific for certain cleavage sites.

According to the dogma, the proteolytic processing of enteroviral polyprotein is thought to be carried out by viral proteases 2A and 3C and the precursor 3CD protein [14,31]. 2A protease makes the first autocatalytic cleavage already during translation between VP1 and itself. The rest of the cleavages are then supposed to be carried out either by 3C or 3CD. Lawson and Semler (1992) showed in their earlier study in vitro that, as P1 precursor was visible already at 25 min during in vitro translation, there was a delay in the processing until 70 min, although 3CD was detected as early as 50 min [32]. In infected cells, according to their metabolic labeling, 3C and 3CD were more active when enriched in the membranous fraction while P1 precursor and its processing occurred very efficiently also in the soluble fraction [32]. Perhaps the P1, which is found close to the replication membranes, are processed by 3C or 3CD, but the P1 found in the soluble fraction may be preferentially processed by calpain proteases readily available in the same pool. Whether calpain proteases truly function in infected cells and take part in polyprotein processes remains to be shown.

Viral proteases and calpains belong to the class of cysteine proteases and are both present in infected cells. Due to difficulties in differentiating calpain from viral protease action, the possible participation of calpains in polyprotein processing is difficult to demonstrate. The use of calpain inhibitors in virus infection assays is challenging due to cross-reactivity with viral proteases 2A and 3C, as has been shown for 2A by others as well [30]. Therefore, we wanted to study the action of calpains separately from viral proteases and to study the direct effect of purified calpain proteases on viral polyprotein processing. We produced the P1 region of CVB1 polyprotein in baculovirus insect cell expression system and incubated it with the purified calpains 1 and 2 in vitro. These studies and the mass spectrometry analysis unequivocally demonstrated that calpains could cleave at the polyprotein between VP3 and VP1 using cleavage sites differing with only two amino acids to that utilized by enteroviral protease 3C. However, further studies are needed to verify if the capsid proteins produced by calpains could be used in the production of pentamers and subsequently in the assembly of infectious virions. Interestingly, it has been shown earlier with foot-and-mouth disease virus that even longer amino acid segments can be attached to VP1 without hindrance in capsid production [33]. They showed that empty capsids and infectious virions were produced even if the cleavage between VP1 and 2A was prevented. This further suggests that the two-amino-acid difference in the cleavage site of calpains compared to 3C does not hinder capsid production. However, this remains to be shown.

Thus, as a proof of principle, we showed that calpains can process enterovirus polyprotein in vitro. Does this happen in cells as well? It has been shown earlier that enteroviruses affect the calcium homeostasis during infection by releasing calcium stores from ER and by increasing the permeability of plasma membrane, resulting in the increase of cytosolic calcium concentration [34,35]. Therefore, changes in calcium homeostasis due to virus infection may trigger calpain activation. Furthermore, Bozym et al. showed that the changes in calcium concentration during CVB3 infection occurred at 2–3 h p.i. [24], and our earlier results showed that calpain activity increased at 3–4 h p.i. [8]. Altogether, these earlier studies supported the idea that calpains get activated after virus entry during the first steps of translation/replication. Furthermore, our earlier studies using calpain-specific siRNAs more specifically showed that calpains themselves were needed for infection [8]. Interestingly, Bozym et al. found less effect on CVB3 infection with calpain siRNAs, which could possibly be due to the ineffectiveness of the siRNAs, leading to only a partial silencing of calpain genes [24]. Also, in our earlier studies, calpain siRNAs decreased EV1 infection but did not block it completely [8]. In addition to calpains 1 and 2, there are 13 other calpains that possibly contribute to the proteolytic processing, thus making it difficult to completely get rid of calpain action in the cells using siRNAs.

Calpain proteases are very abundant in cells, and their activation is controlled by calcium concentration, a process in which enteroviruses also contribute to. In vitro assays have shown that the calcium requirement of calpain 1 and 2 differ: calpain 1 needs micromolar whereas calpain 2 needs millimolar concentrations of calcium [20]. The calpain activation, especially in vivo, is still under investigation to understand how the proteases are activated in cells, where such high calcium concentrations do not exist. However, couple different mechanisms have been suggested: (1) On one hand, increased calcium concentration in the cytoplasm may trigger an autolysis in the N-terminal region of both catalytic and regulatory subunits, resulting in conformational change and finally calpain activation [36,37]. The autolysis may then increase calpain turnover, sensitivity to calcium and substrate accessibility. (2) On the other hand, increased calcium concentration may promote the translocation of calpains to membranes, where the calpains are activated without autolysis in the presence of protein activators and other factors such as phospholipids [38,39,40,41,42]. The presence of activators may decrease the calcium requirement to physiological levels, and translocation to membranes may especially promote membrane-associated proteins as substrates. Additionally, it has been suggested that calpain 1 activation in the presence of micromolar calcium concentration could trigger a cascade where also calpain 2 is activated even with micromolar calcium level [43]. In our in vitro assays, we saw that calpain 2 released more VP1 and VP3 proteins from enterovirus polyprotein compared to calpain 1. However, mass spectrometry analysis showed that both calpains were able to cleave at the VP1-VP3 cleavage site in the same location. The elevated calcium concentrations in cells, which trigger calpain activation, do not resemble the in vitro conditions, and hence, the assessment of calpain isoform preference for enterovirus polyprotein processing from in vitro assays is difficult.

There are a few studies which connect calpain proteases to enterovirus infection after cell entry during trafficking of CVB3-containing vesicles into the perinuclear region [23] for CVB-induced necrosis [24] or for autophagy [25,44]. Most of these studies were carried out using calpain inhibitors, which, as we have shown, might be problematic due to the cross-reactivity with viral proteases. It is difficult to distinguish the effect of the inhibitor on calpains or viral proteases in infection assays, which may confound the outcomes in cellular responses and the interpretation of the results. In addition, there are differences between enterovirus serotypes: we have shown earlier that, in contrast to coxsackie B viruses, EV1 does not need autophagosomes in its infection [8,9]. This was shown using an autophagosomal marker LC3, of which the overexpression did not increase EV1 infection and the amount or size of the LC3 structures did not increase during EV1 infection [8,9]. Thus, our earlier study suggests that, at least during EV1 infection, the role of calpains is not related to autophagy. On the other hand, taking into account the abundancy of calpains in cells and that their activity is mainly regulated by calcium concentration, calpains may play a role in multiple steps during the infection. In addition to enteroviruses, calpains have also been associated with other virus infections, such as influenza, herpesvirus, chikungunya, and other picornaviruses [45,46,47,48,49,50]. Particularly, processing of a viral protein by calpains has been shown with another RNA virus, Hepatitis C virus [51]. Since calpain activity has been shown to be involved in the infections of several viruses, calpains have been suggested as potential targets for antiviral therapies. Our previous and current results suggest that calpain inhibition could also be targeted against enteroviral infections and diseases such as myocarditis, meningitis, or sepsis-like syndrome of infants [2,3,4,52,53,54,55]. It can be speculated that, during evolution, enteroviruses have adapted to using the readily available and active cellular proteases to promote their infection. It may well be that, along time, calpains have become an important backup system in polyprotein processing, while the viral proteases have promoted the infection also by other means, such as cleaving host cell factors related to apoptosis, immune responses, and cellular translation [10]. It has been shown that enteroviruses may stay silent or persistent in tissues for long periods without causing much damage or cell death [3,56,57]. In such circumstances, the viral proteases are likely to be downregulated not to promote apoptosis and cell death, while calpains may still help the polyprotein to be processed further.

The extended P1 polyprotein containing mutated 2A suggested that 2A protease is needed in cis to execute the first cleavage between P2-P3 and P1. This action could not be executed by calpains or by 2A in trans. In addition, the action of 2A could not be inhibited by the specific 2A inhibitor elastatinal during infection of EV1, CVA9, or CVB1 although that inhibitor could inhibit 2A action against the cellular targets in trans. Probably, elastatinal was not able to bind to 2A before 2A was released from the polyprotein and, hence, could not prevent cis action of 2A and, further, the infection. The low inhibitory effect of elastatinal has also been shown with poliovirus earlier [30]. Whether calpains can cleave nonstructural proteins from other 3C/3CD specific cleavage sites at the P2-P3 region remains to be studied.

## 5. Conclusions

Our results demonstrate that calpain proteases can proteolytically process enterovirus polyprotein in vitro, resulting in the release of capsid proteins. Furthermore, the high cross-reactivity of calpain inhibitors with viral proteases observed in this study highlights their potential also as future antivirals.

## Figures and Tables

**Figure 1 viruses-11-01106-f001:**
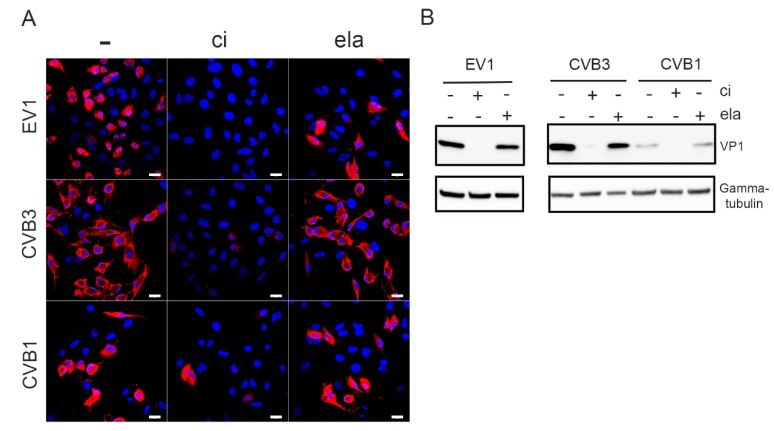
Calpain inhibitor prevents the infection by enteroviruses: A549 cells were infected with EV1, CVB3, or CVB1 for 5.5. h p.i. with or without 200 µM calpain inhibitor (ci) or 250 µM elastatinal (ela). (**A**) Immunofluorescence microscopy was used to evaluate the infection. Viral protein VP1 is in red, and DAPI-stained nuclei are in blue. Scale bar 20 µM. (**B**) For Western blot analysis, the infected A549 cell lysates were resolved in 4–20% SDS-PAGE gels, blotted, and immunolabeled with antibodies against VP1 and gamma-tubulin as a loading control.

**Figure 2 viruses-11-01106-f002:**
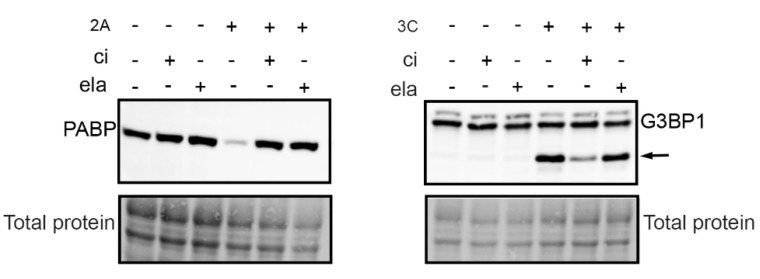
Calpain inhibitor cross-reacts with viral proteases. A549 cell homogenate was incubated with purified viral proteases 2A and 3C in the presence or absence of 200 µM calpain inhibitor (ci) or 250 µM elastatinal (ela) for 18 h at 22 °C. Western blot analysis was done by resolving the samples in 4–20% SDS-PAGE gels and transferring to PVDF membrane. The blots were immunolabeled with PABP and G3BP1 antibodies, and the total protein amount was assessed using a stain-free method. Arrow indicates 3C protease induced cleavage fragment of G3BP1.

**Figure 3 viruses-11-01106-f003:**
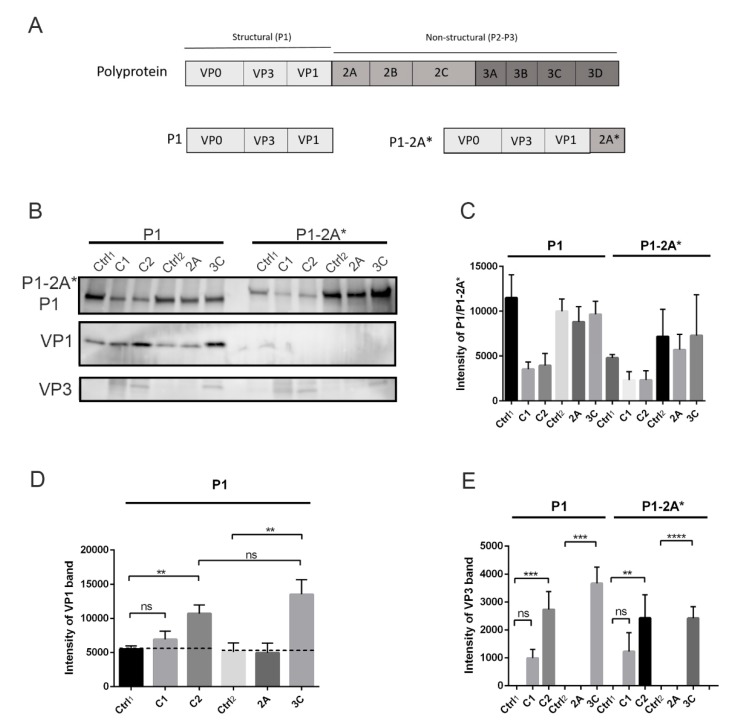
Processing of P1 and P1-2A* constructs by calpains and viral proteases in vitro: P1 and P1-2A* constructs were incubated with purified calpain 1 (C1) or 2 (C2) at +25 °C for 2 h or with viral proteases 2A or 3C at 22 °C for 18 h. Negative controls where no proteases were added were also included: ctrl1 and ctrl2 for calpain and viral protease reactions, respectively. (**A**) Schematic image of P1 and P1-2A* constructs. (**B**) Representative image of P1, P1-2A*, VP1, and VP3 bands revealed by immunolabeling western blots with antibodies against VP1 and VP3. (**C**–**E**) Quantification of P1, P1-2A*, VP1, and VP3 signal from Figure 3B. Data are presented as means ± SEM from three separate experiments. Significance was determined with one-way ANOVA with Bonferroni’s multiple comparisons test (** *p* < 0.01; *** *p* < 0.001; **** *p* < 0.0001; ns, non-significant).

**Figure 4 viruses-11-01106-f004:**
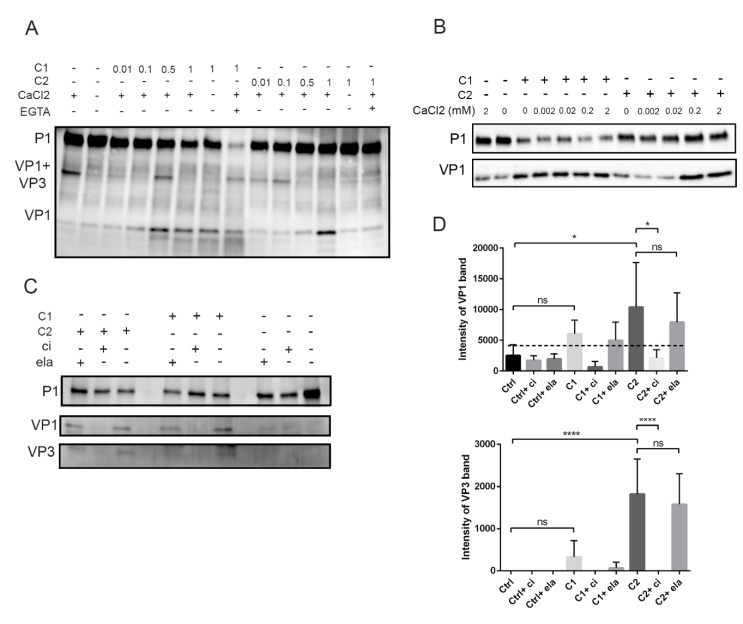
Effect of calpain concentration, calcium, and calpain inhibitor on calpain activity in vitro: (**A**) Calpain 1 (C1) or 2 (C2) were incubated with P1 polyprotein at +25 °C for 2 h. The reactions included 2 mM CaCl_2_ and calpains with increasing catalytic activities: 0.01, 0.1, 0.5, or 1 unit. Also, samples without any proteases or CaCl_2_ or with 4 mM EGTA were included. (**B**) Titration of CaCl_2_ against 1 unit of calpain 1 or 2. The reactions were carried out as in Figure 4A. (**C**) The effect of calpain inhibitor (ci) or elastatinal (ela) on calpain activity during P1 processing: P1 was treated with 1 unit of calpain 1 (C1) or 2 (C2) in the presence or absence of 200 µM calpain inhibitor (ci) or 250 µM elastatinal (ela) for 2 h at +25 °C. Representative images of P1, VP1, and VP3 bands. (**D**) Quantification of the VP1 and VP3 band intensities from Figure 4C: Data are presented as means ± SEM from three separate experiments. Significance was determined with one-way ANOVA with Bonferroni’s multiple comparisons test (* *p* < 0.05; **** *p* < 0.0001; ns, nonsignificant).

**Figure 5 viruses-11-01106-f005:**
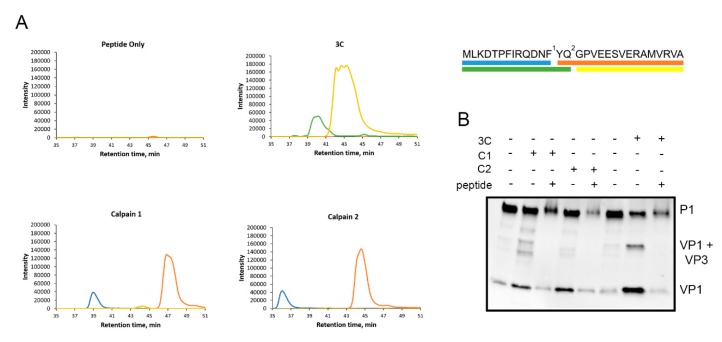
Calpains cleave specifically between VP1 and VP3. A 30-amino-acid artificial peptide around the VP1-VP3 cleavage site was used to study the cleavage site of calpains in more detail. (**A**) The peptide was incubated with calpain 1 or calpain 2 for 2 h at +25 °C or with 3C for 20 h at RT. Also, a control reaction with no proteases was included. The cleavage products were analyzed with SWATH-MS. The peaks correspond to different cleavage products, of which the sequences are color coded in the schematic image of the peptide on the right. Numbers 1 and 2 indicate the cleavage sites of calpains and 3C, respectively. (**B**) P1 construct was incubated with calpain 1 (C1) or 2 (C2) for 2 h at +25 °C or with 750 ng of 3C protease for 17 h at 22 °C with or without 100 µg of the peptide. Also, controls with no proteases were included. Representative images of three experiments are shown.

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
