# Peer review of "Host Cell Calpains Can Cleave Structural Proteins from the Enterovirus Polyprotein"

_viruses, 2019, doi:10.3390/v11121106_

Round 1
Reviewer 1 Report
No other comments for this resubmission
Author Response
We are glad that our revision was approved by referee 1.
Reviewer 2 Report
This revised manuscript from Laajala et al has addressed several of the points I raised in my original review. They have presented new data (figure 5) showing that a peptide competitor inhibits not only calpain 1 and 2, but also 3C, as it should. They have also added a titration analysis to clarify the impact of Ca++ on calpain1 and 2 activity. Lastly, the have confirmed that the presence of a Calpain-like protease in insect cells that is responsible for low level of P1 cleavage seen in purified samples. I would suggest adding a description of this data to the manuscript so that readers can appreciate the reason for the appearance of VP1 and other cleavage products in the absence of exogenously added protease. I don’t think it is necessary to add the figure provided in their response to the reviewer’s comments. Over all the authors have done an excellent job of responding to the original criticisms and the manuscript is significantly improved.
Author Response
We thank for the kind words by reviewer2. We have now added sentences in the results section to point out that a calpain-like protease in insect cells causes a low background in the western blots and that this can be prevented by adding calpain inhibitor in the insect cells cultivation solution.
The added sentences have been pinpointed by underlining.
This manuscript is a resubmission of an earlier submission. The following is a list of the peer review reports and author responses from that submission.
Round 1
Reviewer 1 Report
Prior studies from this group showed that chemical or siRNA-mediated inhibition of calpain proteases block replication of Echovirus 1 and CVB3 although the exact contribution of calpain to viral replication is not known. In this study, the authors examine the possibility that calpains may contribute to cleavage of the viral polyprotein.
Infection of A549 cells with Echovirus 1, CVB1 and CVB3 in the presence of calpain inhibitor 1 (CI) blocks accumulation of the viral VP1 protein assessed by immunofluorescent microscopy and western blotting. Using In vitro assays CI is shown to inhibit the ability of 2A and 3C to proteolyze known cellular substrates. Incubation of P1-2A polyprotein purified from insect cells with purified protease shows that calpain 1 and 2 can cleave the P1 region, releasing a product very similar in size to VP3. Analysis of mutant construct with a catalytically inactive 2A revealed the calpain1/2 don’t cleave at the VP1-2A junction. Using purified P1 the authors show that cleavage by CP1 is independent of Ca++, while that of CP2 is not. Using synthetic peptides spanning the VP3-VP1 3C cleavage site and SWATH-MS the cleavage site for CP1 and 2 was mapped to the P2 position relative to the 3C cleavage site.
General Comments:
The results of this study clearly demonstrate that calpain 1 and 2 can cleave the enterovirus P1 region near sites that are recognized by the viral 3C protease. The experiments demonstrating this are well done and convincing and raise important questions that will be of interest to the field. Since CI also inhibits both 2A and 3C an outstanding question is whether or not cleavage by C1 and C2 contributes to replication of enteroviruses in cells. The results of prior work showing that knockdown of C1/C2 impacts replication suggests that it might, however this question is not pursued further in this study. Another important question raised by the study, and commented on in the discussion, is if the variants of VP1, VP3 (and possibly VP0) produced by CP1/2 proteolysis exhibit any difference in function from those produced by 3C cleavage.
Specific Comments:
Has the last author been left off the author list?
Figure 1- The authors state that elastatinal did not suppress infection. This is surprising given its known ability to inhibit 2A (as demonstrated in Fig 2). In addition, it looks like VP1 levels are down slightly, at least for EV1 and CVB3. Consequently, it would be useful to evaluate the impact of these inhibitors on viral replication in a more quantitative way. For example, is it possible to quantitate the amount of fluorescent signal? Or assay the impact of CI and Ela on viral titers?
Figure 3/4- It appears that the Substrate (P1-2A or P1) is proteolyzed in the absence of purified proteases giving rise to VP1 and VP1/3. of VP1 in the absence of any protease, but only when Ca++ is added. Have the authors examined if CI block the appearance of these cleavage products?
Figure 5- It would be interesting to know if the competitor peptide inhibits 3C cleavage of P1 more or less efficiently than that of C1 and C2.
Author Response
Prior studies from this group showed that chemical or siRNA-mediated inhibition of calpain proteases block replication of Echovirus 1 and CVB3 although the exact contribution of calpain to viral replication is not known. In this study, the authors examine the possibility that calpains may contribute to cleavage of the viral polyprotein.
Infection of A549 cells with Echovirus 1, CVB1 and CVB3 in the presence of calpain inhibitor 1 (CI) blocks accumulation of the viral VP1 protein assessed by immunofluorescent microscopy and western blotting. Using In vitro assays CI is shown to inhibit the ability of 2A and 3C to proteolyze known cellular substrates. Incubation of P1-2A polyprotein purified from insect cells with purified protease shows that calpain 1 and 2 can cleave the P1 region, releasing a product very similar in size to VP3. Analysis of mutant construct with a catalytically inactive 2A revealed the calpain1/2 don’t cleave at the VP1-2A junction. Using purified P1 the authors show that cleavage by CP1 is independent of Ca++, while that of CP2 is not. Using synthetic peptides spanning the VP3-VP1 3C cleavage site and SWATH-MS the cleavage site for CP1 and 2 was mapped to the P2 position relative to the 3C cleavage site.
General Comments:
The results of this study clearly demonstrate that calpain 1 and 2 can cleave the enterovirus P1 region near sites that are recognized by the viral 3C protease. The experiments demonstrating this are well done and convincing and raise important questions that will be of interest to the field. Since CI also inhibits both 2A and 3C an outstanding question is whether or not cleavage by C1 and C2 contributes to replication of enteroviruses in cells. The results of prior work showing that knockdown of C1/C2 impacts replication suggests that it might, however this question is not pursued further in this study. Another important question raised by the study, and commented on in the discussion, is if the variants of VP1, VP3 (and possibly VP0) produced by CP1/2 proteolysis exhibit any difference in function from those produced by 3C cleavage.
We thank the referee for the kind comments and for seeing the importance of our study.
Specific Comments:
Has the last author been left off the author list?
The editorial office has added ‘’and’’ after Varpu Marjomäki for some reason. There are no additional names coming after Marjomäki. All authors are listed in the author list. This has been corrected in the revised manuscript.
Figure 1- The authors state that elastatinal did not suppress infection. This is surprising given its known ability to inhibit 2A (as demonstrated in Fig 2). In addition, it looks like VP1 levels are down slightly, at least for EV1 and CVB3. Consequently, it would be useful to evaluate the impact of these inhibitors on viral replication in a more quantitative way. For example, is it possible to quantitate the amount of fluorescent signal? Or assay the impact of CI and Ela on viral titers?
We thank the referee for good comments. We believe that elastatinal did not inhibit the cleavage action of 2A when it was still attached to the polyprotein, but was able to inhibit 2A action in trans. It is possible that in cis, the binding of 2A to the polyprotein motif cannot be disturbed by exogenous drugs due to steric reasons, but in trans, the drugs have a better access to interfere with binding. In addition, when 2A is attached to the polyprotein the local concentration of the protease is high in relation to the target, and thus, the concentration of elastatinal needs to be high for efficient inhibiton. However, it has also been shown earlier with poliovirus that elastatinal, even at higher concentrations has only a minor effect on infection (Molla et al. 1993; ref.30). We have added discussion on this matter (lines 504-506). Therefore, we think that quantification is not necessary here as it would likely show very low effect. We have shown the negative effect of elastatinal already before with increasing series of elastatinal added (Upla et al. 2008; ref 8). In the same article, the effect of calpain inhibitor on enterovirus B infection and replication was already carefully quantified (Upla 2008; ref 8). The results showed that the inhibitor prevented the protein production and replication of EV1 detected by immunolabeling of VP1 protein and quantifying the amount of both (+) and (-) strand RNA using quantitative RT-PCR, respectively.
Figure 3/4- It appears that the Substrate (P1-2A or P1) is proteolyzed in the absence of purified proteases giving rise to VP1 and VP1/3. of VP1 in the absence of any protease, but only when Ca++ is added. Have the authors examined if CI block the appearance of these cleavage products?
Referee is right. We see ‘’background’’ of VP1 and VP1/VP3 in the control samples. This proves us that there are endogenous proteases in insect cells which are similar enough to the added calpains to execute the cut, although in low efficiency. Nevertheless, exogenous purified calpain proteases could process P1 and release VP1 and VP3 proteins at higher levels compared to the background. Exogenously added calpain inhibitor does not erase the already processed VP1 from the background, but, in our assays, addition of calpain inhibitor during P1 production with baculovirus system and monitoring VP1 appearance shows a decrease of the background signal (Fig.1). After two days post-infection, P1 was produced both with and without calpain inhibitor but VP1 was released from P1 only in the control production. Thus, calpain inhibitor prevented the proteolytic processing carried out by calpain or similar protease in the insect cells.
Please see the attached Fig.1.
Fig.1. Processing of recombinant enterovirus polyprotein segment corresponding to structural proteins (P1) prevented by calpain inhibitor in insect cells. P1 was produced with baculovirus expression system in sf9 cells for 2 days. Calpain inhibitor (ci) was added into the culture medium between 1 and 2 days post infection (dpi) and also control P1 production with no inhibitor was included (ctrl). Western blot was immunolabeled with Enterovirus clone 5-D8/1 antibody to reveal P1 and VP1.
Figure 5- It would be interesting to know if the competitor peptide inhibits 3C cleavage of P1 more or less efficiently than that of C1 and C2.
We thank the referee for a good suggestion. Now we have added a new figure in 5B where we show that the presence of the peptide prevents the action of both calpains and 3C protease in a similar manner.
Reviewer 2 Report
Because the authors provided the results shown in Fig. 4 and revealed that activity of calpain 1 and 2 are calcium-dependent. Also, authors mentioned that the calpain 1 needs micromolar, whereas calpain 2 needs milliomolar level of calcium. Therefore, authors should test the different concentration of CaCl2 in the in vitro assay as following the procedure of Fig. 4. to investigate the differential requirement of calcium concentration in between calpain 1 and calpain 2 during the cleavage of VP1-VP3 of enterovirus.
Author Response
Because the authors provided the results shown in Fig. 4 and revealed that activity of calpain 1 and 2 are calcium-dependent. Also, authors mentioned that the calpain 1 needs micromolar, whereas calpain 2 needs milliomolar level of calcium. Therefore, authors should test the different concentration of CaCl2 in the in vitro assay as following the procedure of Fig. 4. to investigate the differential requirement of calcium concentration in between calpain 1 and calpain 2 during the cleavage of VP1-VP3 of enterovirus.
We thank the referee for a good suggestion. We have now carried out a calcium titration assay where we show that calpain 2 requires higher amount of calcium compared to calpain 1. This has been added in the figure 4B.